# Geochemical and Mineral Properties of Quaternary Deep-Sea Sediments in the Central-Tropical Pacific and Its Response to the Mid-Pleistocene Transition

**Haifeng Wang** [1,2], **Liang Yi** [3,*], **Xiguang Deng** [1,2] **and Gaowen He** [1,2]

1   Southern Marine Science and Engineering Guangdong Laboratory (Guangzhou), Guangzhou 511458, China; wanghaifeng@mail.cgs.gov.cn (H.W.); dxiguang@mail.cgs.gov.cn (X.D.); hgaowen@mail.cgs.gov.cn (G.H.)
2   Key Laboratory of Marine Resources, Ministry of Nature Resources, Guangzhou Marine Geological Survey, China Geological Survey, Guangzhou 510075, China
3   State Key Laboratory of Marine Geology, Tongji University, Shanghai 200092, China
*   Correspondence: yiliang@tongji.edu.cn

**Abstract:** Global climate and oceanic water masses have undergone profound changes during the middle Pleistocene transition; however, due to a lack of foraminiferal fossils, the nonfossiliferous pelagic deposits were less detected in previous reports. In this work, a gravity core from the Kamehameha Basin in the Central Pacific was studied in terms of magnetostratigraphy, clay mineral and geochemical elements. The main results are: (1) nine magnetozones are recognized in the core, which can be correlated to the geomagnetic polarity timescale from chrons C2n to C1n; (2) smectite is the dominant clay mineral, and the others are illite, chlorite and kaolinite; and (3) the sediments are mainly composed of $Al_2O_3$, $Fe_2O_3$, $MnO$, $Na_2O$ and $TiO_2$. Based on these results, a geochronological framework for the study area was established, and the depositional rates are estimated as 3–7 m/Myr in the Quaternary, showing an increase during the middle Pleistocene transition. By comparing the findings to various paleoenvironmental processes, it is inferred that the increased sedimentation in the Kamehameha Basin may have resulted from the induced weathering processes and the strengthened aeolian inputs from inner Asia. Moreover, regional circulation related to bottom water evolution has experienced a rapid reorganization across the middle Pleistocene transition. All these findings illustrate the potential of deep-sea sediments in the central tropical Pacific in revealing some key features in paleoclimatology and paleoceanography, which are worthy of further investigation in the future.

**Keywords:** magnetostratigraphy; clay minerals; geochemical properties; deep-sea sediments; mid-Pleistocene transition

## 1. Introduction

The past climate was dominated by a symmetric 41 kyr cycle in the early Pleistocene, and by an asymmetric 100 kyr cycle since the middle Pleistocene [1], and this change in dominant cycles is the so-called Middle Pleistocene transition (MPT, ~1.45–0.9 Ma). Over the past decades, many efforts have been made to reveal the characteristics and mechanism of the MPT [2–4], and a major shift at ~900 ka was commonly found in paleoclimatic records, such as the LR04 $\delta^{18}O$ isotopic stack [5], ice-rafted debris into the Norwegian Sea [6], North Atlantic sea surface temperatures (SSTs) at DSDP Site 607 [7] and Eastern Tropical Atlantic SSTs at ODP Site 1077 off the west coast of Africa [8]. However, deep-sea sedimentary processes below the calcite compensation depth (CCD) responding to the MPT are not yet well documented.

The pelagic sediments in the tropical Pacific may have fully recorded changes in paleoceanography, provenance and global climate, and their characteristics can be employed to reveal such information [9].

The Kamehameha Basin connects the Central Pacific and the Eastern Pacific (Figure 1). The Lower Circumpolar deep water (LCDW) and the Antarctic bottom water (AABW) flowed into the open Eastern Pacific Basin through the Horizon and Clarion Passages [10]. This region is an area with significant interactions between various systems: (1) the south is the high-productivity equatorial Pacific belt, and the north is controlled by the North Pacific Gyre [11]; (2) the sediments contain biogenic debris and aeolian dust; (3) volcanogenic materials are evident from the Hawaii hotspot [12]; and (4) fresh volcanic ashes are commonly found. Thus, sedimentary properties of the deep-sea sediments in the Kamehameha Basin may record the interactions between various climatic and oceanic systems.

However, such studies on the nonfossiliferous sediments were less studied in previous research due to age uncertainties [11,13], limiting our understanding of how deep ocean was affected during global changes. In this work, we studied a gravity core PC15 collected from the Kamehameha Basin. Combining the results of magnetostratigraphy, clay minerals and geochemistry, the properties of the deep-sea sediments were well studied in the Quaternary. By comparing the results of the core with various paleoenvironmental proxies during the MPT, we attempt to study (1) the sedimentary processes, (2) changes in the provenance, and (3) bottom water evolution in the study area.

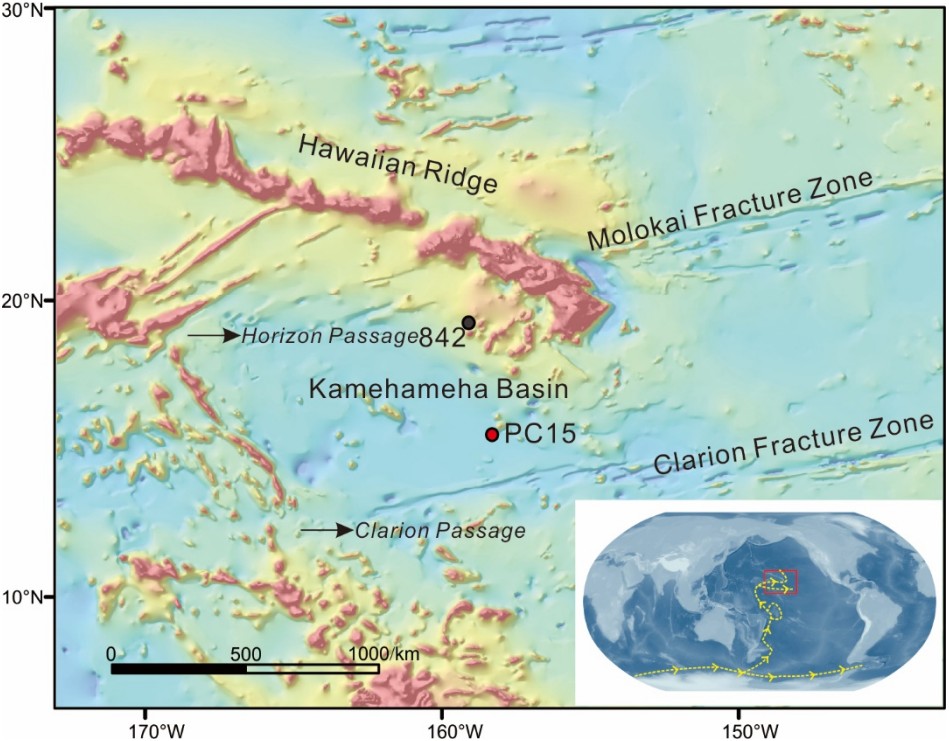

**Figure 1.** Schematic map of the central Pacific region showing the study sites and other referenced sites. The yellow dashed line in the bottom right insert shows the paths of the Lower Circumpolar deep water (LCDW) and the Antarctic Bottom Water (AABW). The base data are from ref. [14] and generated using the open and free software DIVA–GIS 7.5 (http://www.diva-gis.org/, accessed on 10 September 2021).

## 2. Materials and Methods

### 2.1. The Studied Core

Core PC15, located between the Horizon Passage and the Clarion Passage (158.25° W, 15.53° N, 5525 m water depth), was collected in 2013 by the Guangzhou Marine Geological Survey with a length of 5.65 m (Figure 1). The oceanographic setting of the studied area is dominated by the LCDW and the AABW [11,15]. Since the locality is generally below the CCD (~4500 m [16]), the core mainly contains homogenous carbonate-free muds.

According to the lithological description (Figure 2), core PC15 can be divided into two units around 0.92 m: the upper Unit I contains yellow muds (10YR7/4) with black laminates with volcanic ashes, and the lower Unit II contains clay in pale brown (10YR8/2) and clay with zeolite in dark brown (10YR3/3).

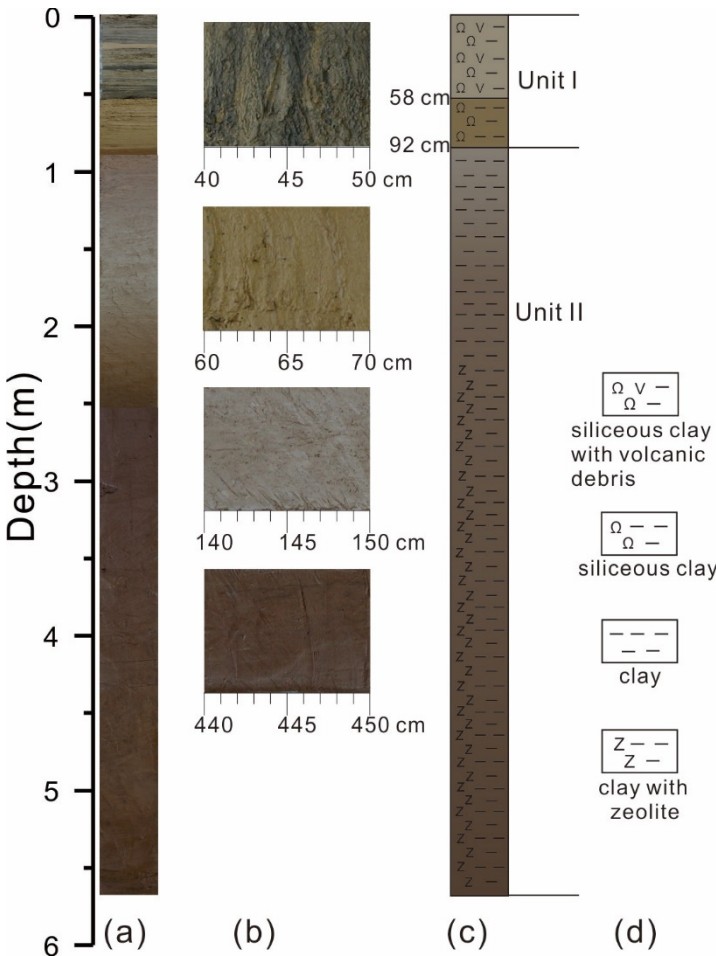

**Figure 2.** Photography (**a**,**b**) and lithological changes (**c**) of core PC15; (**d**) lithological legends.

### 2.2. Demagnetization

Paleomagnetic samples were continuously collected using nonmagnetic plastic cubic boxes (2 cm × 2 cm × 2 cm). In total, 282 samples were obtained and subject to stepwise alternating field (AF) demagnetization up to a peak field of 90 mT (13 steps). The natural remanent magnetization (NRM) was measured using a three-axis cryogenic magnetometer (2G Enterprise Model 755-4K, USA) installed in magnetically shielded room (residual fields <300 nT) at the State Key Laboratory of Marine Geology, Tongji University. Characteristic remanent magnetization (ChRM) directions were determined using principal component analysis [17] implemented by the PuffinPlot package [18], with at least four consecutive demagnetization steps.

### 2.3. Clay Minerals Analysis

A total of 28 samples were taken for clay mineral analysis by X-ray diffraction (XRD) at a sampling interval of 20 cm continuously from the top to the bottom of core PC15. The XRD clay mineral study was carried out on the <2 μm fraction, which was separated by conventional Stokes' settling after the removal of carbonate and organic matter by acetic acid (15%) and hydrogen peroxide (10%), respectively. Clay minerals were then identified by XRD using a (Rigaku) D/Max 2500PC 18 kW powder diffractometer (SYM125) in the Guangzhou Marine Geological Survey, with an X-ray wavelength λ = 1.5418Å

(CuKα) and a scanning rate of 2°(2θ)/min at 0.02°(2θ). Each sample was measured 3 times under conditions of air-drying, ethylene glycol solvation, and heating at 490 °C for 2 h at atmospheric pressure [19]. Clay minerals were identified according to the position of the (001) series of basal reflections observed on the XRD diagrams [19,20].

### 2.4. Element Analysis

For trace element content analysis, 28 bulk sediments were completely dissolved by HF and $HNO_3$ solutions. A sample of about 50 mg was weighed and transferred into a pre-cleaned Teflon beaker followed by the addition of ultra-pure 1 mL of HF and 1 mL of $HNO_3$ solution. The beakers were then placed in steel cans and subjected to high temperature (185 °C) and high pressure. After 36 h, the solution was dried on a hotplate. The residues were fully digested using a mixture of concentrated 2 mL $HNO_3$ and 3 mL Milli-Q water. Thereafter, the beakers were placed into steel cans at 120 °C for 5 h. After cooling, the solution was diluted to 20 mL with Milli-Q water. Major elements were analyzed with ICP-OES (Optima 8300, PerkinElmer, MA, USA), and trace elements were measured via ICP-MS (X Series2, Thermo Fisher Scientific, MA, USA). All the samples were analyzed at Guangzhou Marine Geological Survey. Certified reference material (GBW07315) was used for quality control. Precision and accuracy were both better than 5%.

## 3. Results

### 3.1. Magnetostratigraphic Results

In general, the remanence gradually decreases subject to AF demagnetization. About 50% of the NRMs was removed at ~25 mT, and up to a peak field of 80 mT, more than 90% of the remanence was removed. The normal inclinations are from 5° to 40°, and the reverse ones from −8° to −45°. The characteristics of AF stepwise of the representative samples were displayed in the form of orthogonal diagrams (Figure 3).

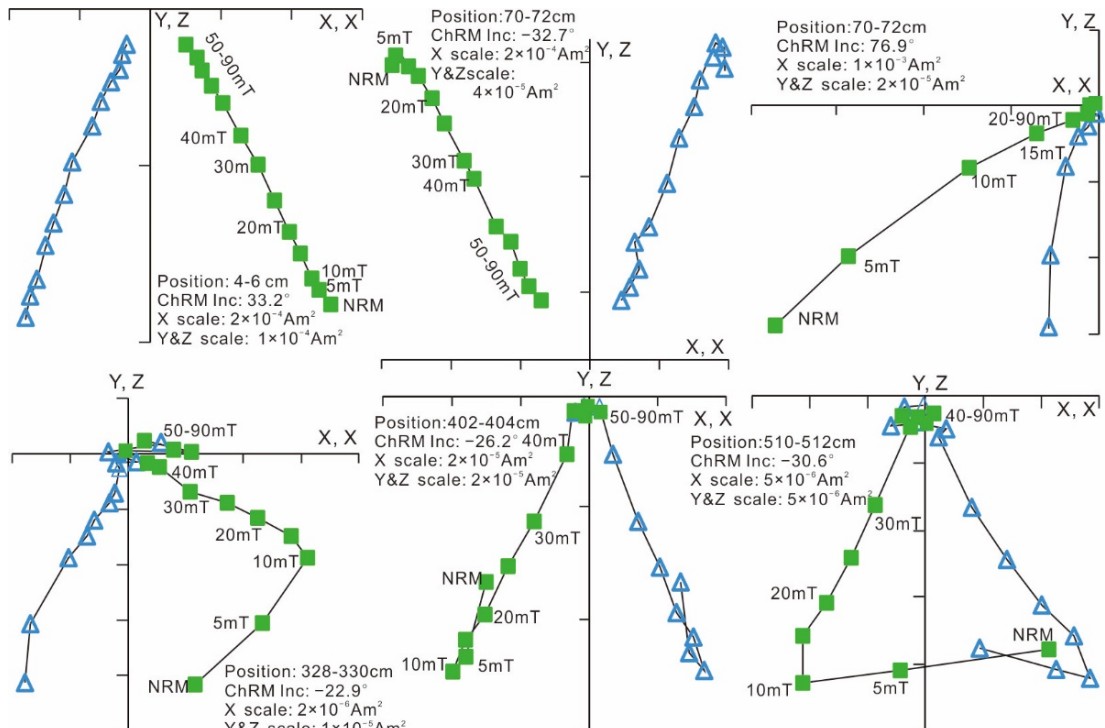

**Figure 3.** Orthogonal diagrams of the stepwise AF demagnetization of representative samples. Open triangles represent the horizontal planes, while solid squares represent vertical planes.

According to the obtained ChRMs, five normal (n1–n5) and four reverse magnetozones (r1–r4) were recognized. Considering the homogeneous sedimentary lithology

and taking the results at ODP Site 842B as a reference [13], the magnetozones can be correlated to the geomagnetic polarity time scale (GPTS) [21]. As particularly shown in Figure 4, the normal magnetozones n1, n2, n3, n4 and n5 are correlate to Chron C1n (Brunhes, 0–0.781 Ma), C1r.1n subchron (Jaramillo, 0.988–1.072 Ma), Cobb Mountain sub-chron (Cobb Mtn., 186–1.221 Ma), the Gilsa excursion (~1.68 Ma) and Chron C2n (Olduvia, 1.778–1.945 Ma), respectively. Subsequently, reversal magentozones can be correlated to the reversed subchrons belonging to the Matuyama Chron. Based on these correlations (Table 1), the sediment accumulation rates (SAR) of core PC15 are estimated as 3–6 m/Myr.

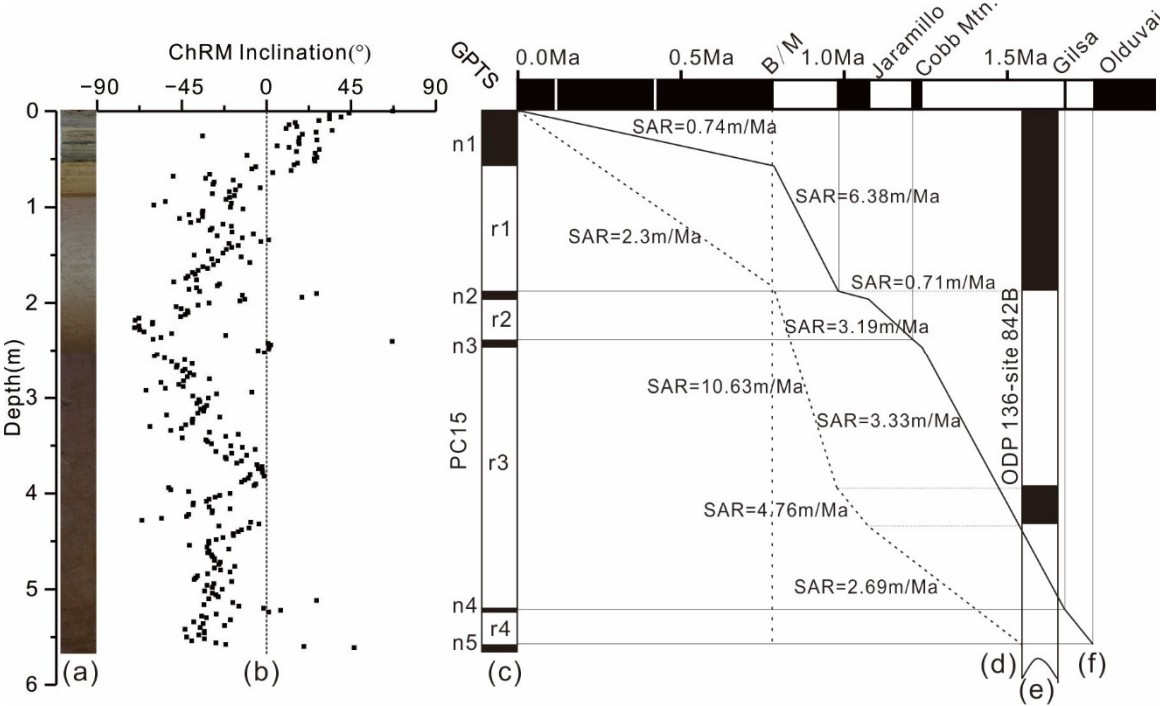

**Figure 4.** Magnetostratigraphy of the studied core and their correlations to the geomagnetic polarity time scale (GPTS). (**a**) Image of Core PC15; (**b**) ChRM inclination; (**c**) recognized magnetozones (n1–n5 and r1–r4) of core PC15; (**d**) sedimentation rates of ODP 842B site (dash line); (**e**) magnetostratigraphic results of core ODP 136-site 842B [13]; (**f**) sedimentation rates of core PC15 (solid line). See data in Table 1.

**Table 1.** Correlation of magnetozones of core PC15 to the Geomagnetic Polarity Time Scale.

| Geomagnetic Polarity Time Scale | Age [21] (Ma) | PC15 Depth (m) | PC15 SARs (m/Ma) | 842 [22] Depth (m) | 842 [22] SARs (m/Ma) |
|---|---|---|---|---|---|
| Surface | 0 | 0 | - | 0 | - |
| C1n (bottom) | 0.781 | 0.58 | 0.74 | 1.8 | 2.3 |
| C1r.1n (top) | 0.988 | 1.9 | 6.38 | 4 | 10.63 |
| C1r.1n (bottom) | 1.072 | 1.96 | 0.71 | 4.4 | 4.76 |
| Cobb Mountain (top) | 1.21 | 2.4 | 3.19 | - | - |
| Cobb Mountain (bottom) | 1.24 | 2.5 | 3.33 | - | - |
| C2n(top) | 1.778 | 5.6 | 5.76 | 6.3 | 2.69 |

Since the sediments of core PC15 deposited below the CCD, most calcareous foraminifera and coccolith have been completely dissolved, and only some pieces of radiolarian fossils are observed. In the topmost 50 cm, diatom fossils were well-preserved, while in the lower, no diatoms were found. The identified fossils include *Azpeitia nodulifera*, *Hemidiscus cuneiformis*, *Azpeitia Africana*, *Nitzschia marina*, *Thalassiosira oestrupii* and *Coscinodiscus excentricus*, supporting the reliability of paleomagnetic correlation (Table 1).



*3.2. Clay Mineral Changes*

Clay minerals in core PC15 are smectite (46–89%) in dominance, with illite (~38%), chlorite (4–11%) and kaolinite (4–8%). High percentages of smectite in clay minerals usually demonstrates that the sediments are closely related to volcano glasses, while illite and chlorite are from physical weathering and kaolinite can indicate warm and humid climates, which is strongly controlled by continental hydrolysis [23–27]. The assemblage of clay minerals of Unit II is close to those of surface sediments in West Philippine Basin [24–26], while Unit I is close to those of sediments of Mariana Trough [26] and Luzon Island [27], as displayed in Figure 5.

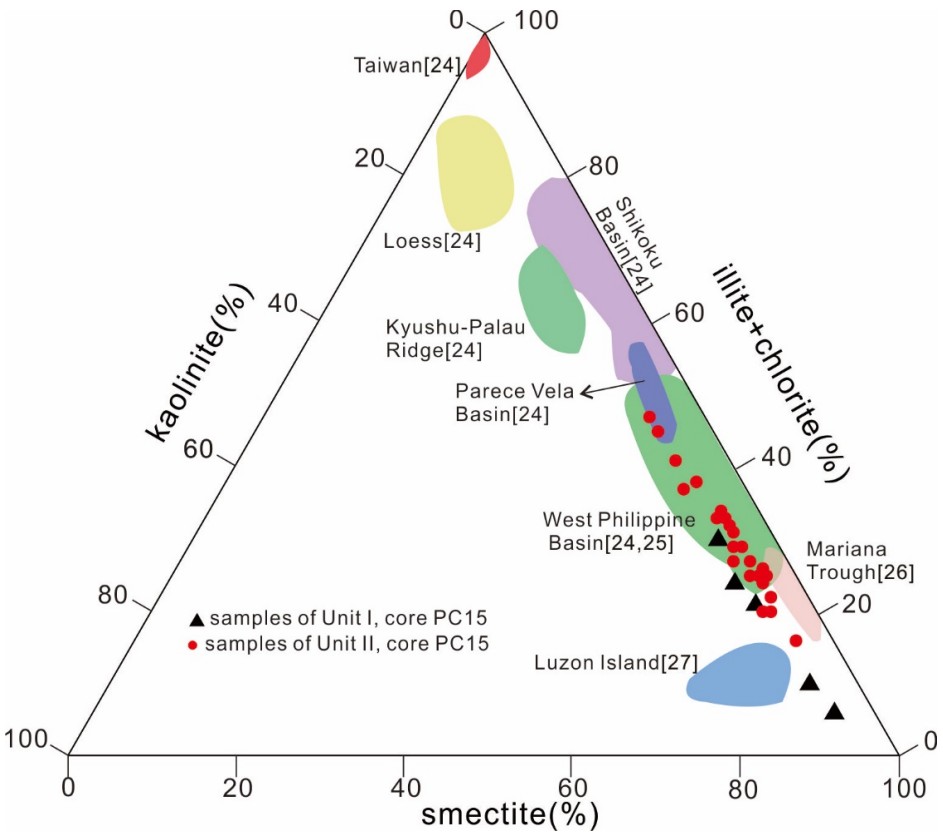

**Figure 5.** Triangular plot of clay minerals' composition.

Illite is the dominant clay mineral in Chinese loesses, up to 65%, and chlorite, ~20%. Kaolinite was commonly found in loesses, about 10%, while smectite was only ~5% [22]. In the West Philippine Basin, changes in clay minerals are large: smectite is high with content of 60–80%, and illite and chlorite range from 16% to 37% [24,25]. In the sediments of the Mariana Trough, more smectite was reported due to regional hydrothermal and volcanic activities [26].

Although the Kamehameha Basin is distant from the Philippine Sea, the provenance between them is similar [28,29]. For example, samples from Unit II of core PC15 displace in the area of the triangular plot similar to ones of the West Philippine Basin and the Mariana Trough, while samples from Unit I are close to samples from the Mariana Trough and the Luzon Islands, which can be correlated to the increased contribution of hydrothermal materials and volcanic debris [13]. Thus, the comparison between clay minerals indicates that aeolian deposits in the sediments generally decrease along with the distance away from the Chinese loess plateau, and hydrothermal materials and volcanic debris significantly increase. For stratigraphic changes, smectite content in the top 0.92 m (Unit I) is higher than the lower part (Unit II), illite reduces from ~30% to a much lower level and changes in chlorite and kaolinite are small (Figure 6).

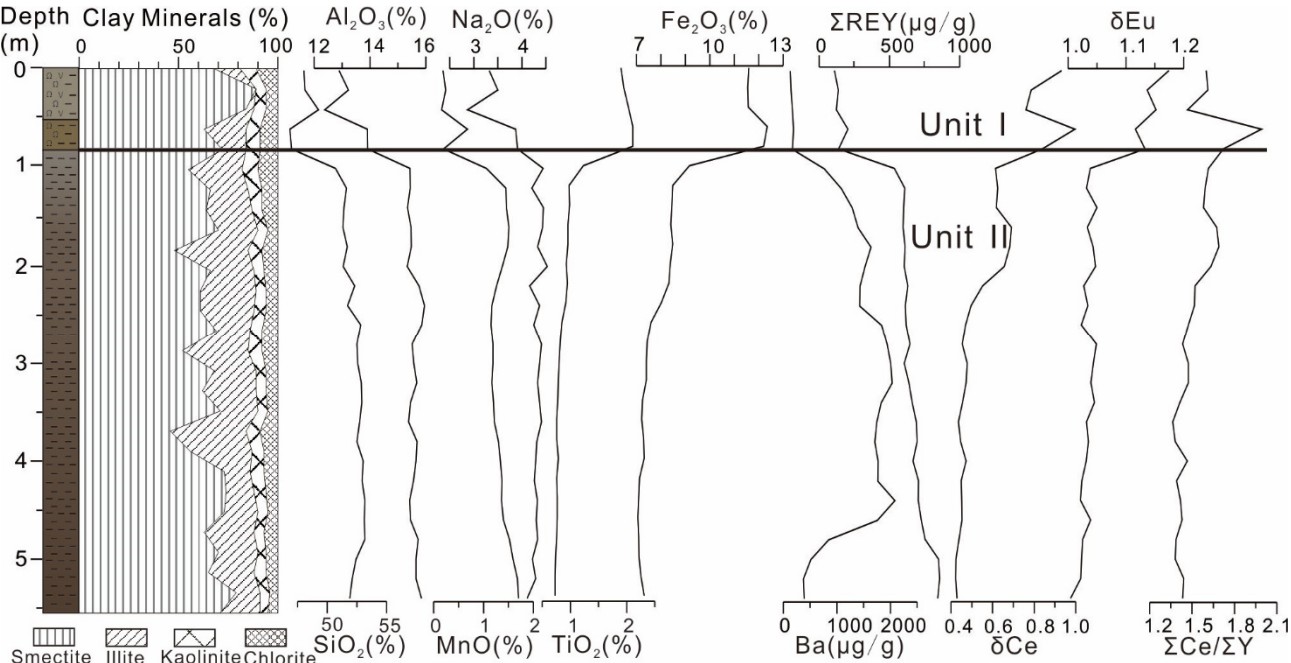

**Figure 6.** Stratigraphic changes in clay minerals, and major, trace and rare-earth elements in core PC15.

### 3.3. Geochemical Properties

Following equations of $\Sigma Ce = La + Ce + Pr + Nd + Sm + Eu$; $\Sigma Y = Gd + Tb + Dy + Ho + Er + Tm + Yb + Lu + Y$; $\delta Ce = 2Ce_N/(La_N + Pr_N)$ and $\delta Eu = 2Eu_N/(Sm_N + Gd_N)$, and normalized by the North American Shale Composition (NASC) [30], geochemical properties of core PC-15 are displayed. The difference between the major elements of $SiO_2$, $Al_2O_3$, MnO and $Na_2O$ is minor, while $TiO_2$, $Fe_2O_3$, $\Sigma Ce/\Sigma Y$, $\delta Ce$ and $\delta Eu$ change in an opposite pattern (Figure 6). Their down-core variation can be divided into two groups: (1) below 0.92 m, all elements change little; and (2) above 0.92 m, $SiO_2$, $Al_2O_3$, MnO and $Na_2O$ gradually decrease, and $TiO_2$, $Fe_2O_3$, $\Sigma Ce/\Sigma Y$, $\delta Ce$ and $\delta Eu$ increase. The detail geochemical properties are listed in Supplementary Tables S1 and S2.

The REE data of core PC15 can be clustered into two parts (Figure 7): (1) in the lower part (Unit II, below 0.92 m), $\Sigma REY$ range from 536 to 860 µg/g, with a distinct Ce negative anomaly; and (2) in Unit I (above 0.92 m), $\Sigma REY$ is less than 200 µg/g, with a slightly positive anomaly in element Eu.

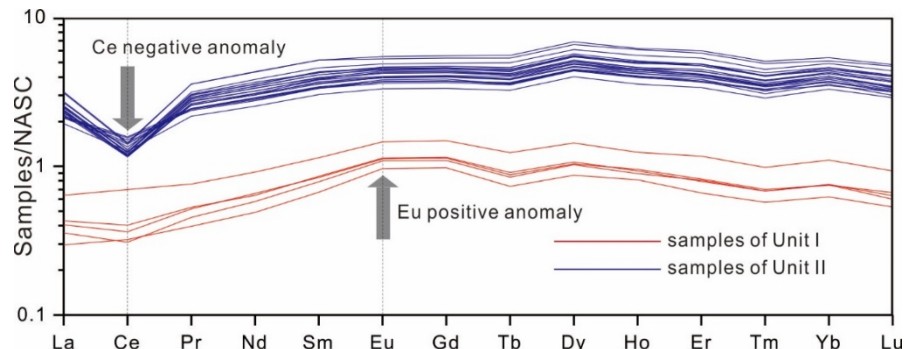

**Figure 7.** The NASC-normalized REE patterns of sediments from core PC15.

## 4. Discussion

### 4.1. Sedimentary Changes during the MPT

Within the MPT, SARs of core PC15 and ODP Site 842 [13] significantly increased, from 3 m/Ma to 6.38 m/Ma in core PC15, and from 4.76 m/Ma to 10.63 m/Ma at ODP

Site 842 [13]. This increase in SARs is not only evident in the Kamehameha Basin, but also in the southern slope of Mariana Trench [14], the Western Pacific and the Philippian Sea [31,32] (Figure 8).

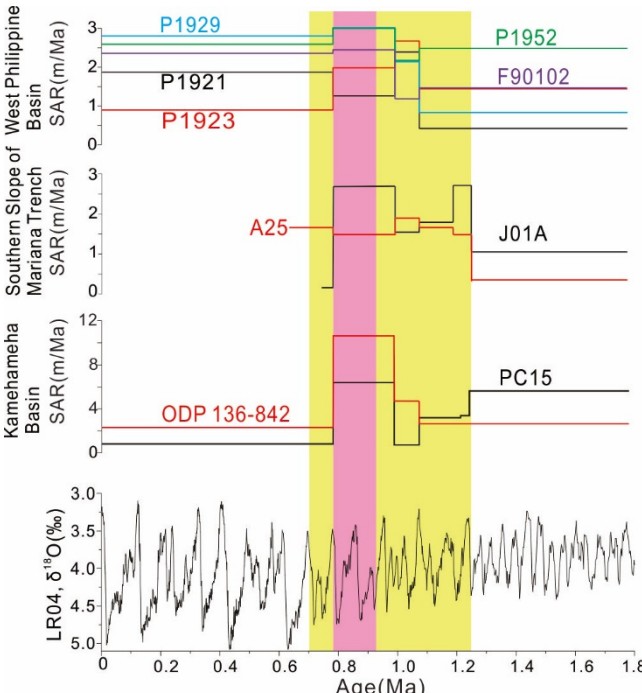

**Figure 8.** Changes in SARs of nine cores from the Kamehameha basin, the southern slope of Mariana Trench and the West Philippine Basin. Benthic $\delta^{18}O$ stack LR04 [6] indicates global climate changes from 41 kyr to 100 kyr world. The Mid-Pleistocene Transition is highlighted by yellow shades, the 900 kyr event with pink shades. ODP Site 842 is from Ref. [13]. Sites A25 and J01A are from Ref. [14]. Sites P1921, P1923, P1929 and P1952 are from Ref. [31]. Site F90102 is from Ref. [32].

During the MPT, there is a major shift in many geological records, such as the LR04 $\delta^{18}O$ stack [5], ice-rafted debris in the Norwegian Sea [6] and changes in SST in the Eastern Equatorial Pacific of ODP 849 Site [33]. Similarly, the mean grain size of loess sequences in the Chinese Loess Plateau increased significantly, inferring a strengthened aridity [34]. Therefore, the increased SAR from the early to middle Pleistocene in the Western Pacific can be explained as follows. The long-term average ice volume gradually increased, about 50 m down in sea-level equivalent [35]. The decreased sea levels would cause more exposure of bedrocks in glacial intervals, and induce weathering processes [34,36]. As a result, the fine-grained weathering particles were carried by regional currents and the Westerlies jet to the study area. Additionally, since biological productivity in the equatorial Pacific Ocean with high nutrients and low chlorophyll was limited by the micronutrient Fe, dust delivery may have induced marine primary productivity by aeolian Fe inputs [37], thus increasing the SAR in the study area.

In addition, since volcanic debris and hydrothermal materials, namely volcanic glasses and smectite, are remarkably higher in Unit I of core PC15, the low SARs in the Kamehameha Basin [13] after the MPT may suggest a substantial reduction in aeolian dusts and biological materials (mainly diatom).

### 4.2. Bottom-Water Evolution

It has been suggested that Ce can be oxidized to $Ce^{4+}$ hydroxide in an oxidized environment due to hydroxide precipitation [38], and negative Ce anomalies are usually observed in the present oxidized deep seawater [38]. On the contrary, deep-sea ferromanganese nodules and crusts usually exhibit positive Ce anomalies, likely reflecting

preferential removal of $Ce^{4+}$ from sea water. Hence, Ce anomaly is a good indicator of sedimentary redox conditions. Moreover, element europium (Eu) is usually in $Eu^{2+}$ in a reducing condition [39]. In high-productivity sea waters, such as the equatorial Pacific and Atlantic water masses, barites ($BaSO_4$) can indicate the upper marine productivity [40–43].

In this study, changes in δCe of core PC15 increased rapidly during 0.9–0.7 Ma (Figure 9), inferring that the bottom current changed from oxic to anoxic conditions. The increased δEu since 0.7 Ma, together with Ba and P components, indicates the ascending influence of marine production in the upper ocean onto the redox condition of bottom water. In addition, the positive Eu anomaly was observed in the top 0.92 m of core PC15, inferring that volcanic materials are an important component in the sediments.

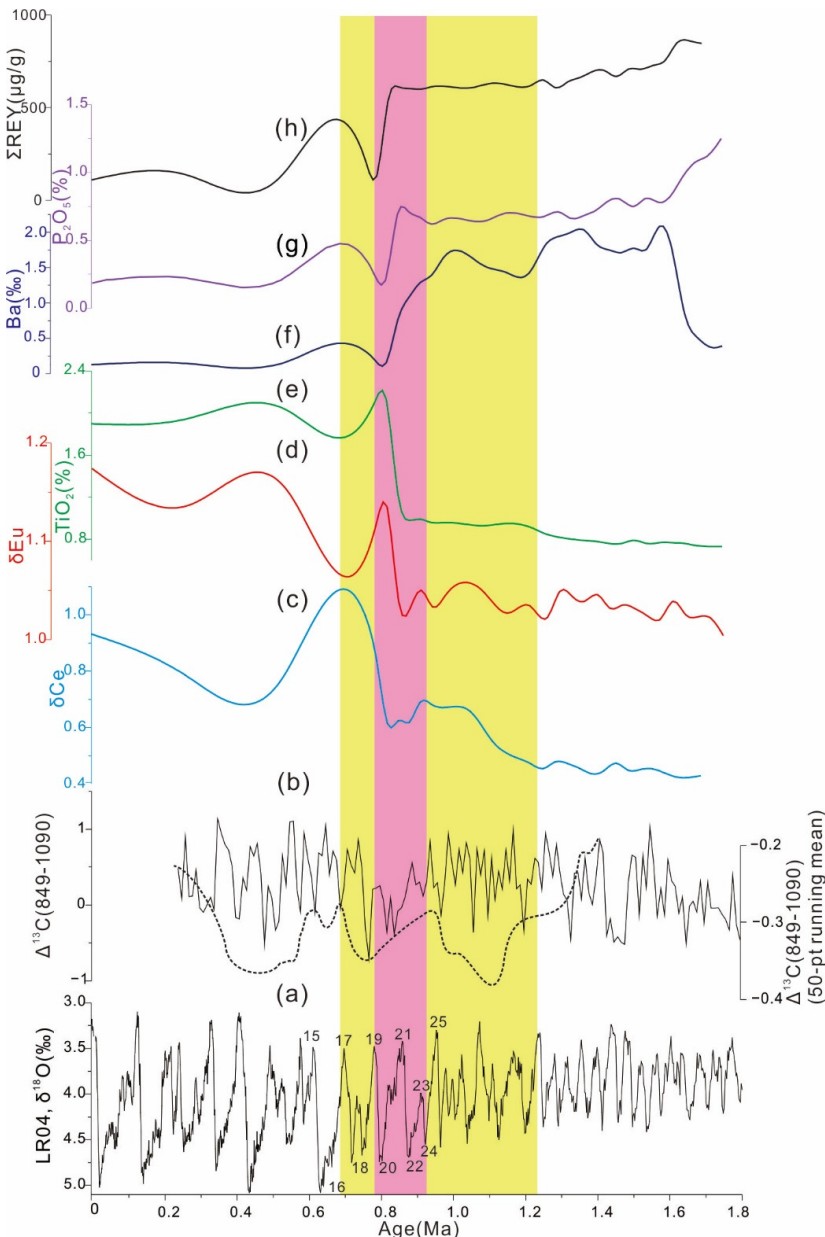

**Figure 9.** Comparison of selected climatic and environmental proxies of sediments from core PC15. (**a**) LR04 marine foraminifer oxygen isotope stack [5]. Marine isotopic stages (15–25) are labeled. (**b**) Inter-basinal carbon isotope gradient (solid line) between Pacific Site 849 and South Atlantic Site 1090 [44], with a 50-point running average (dash line). (**c–h**) δCe, δEu, contents of $TiO_2$ (%), Ba (‰), $P_2O_5$ (%) and REY (μg/g) of core PC15. The Mid-Pleistocene Transition is highlighted by yellow shades and labeled, the 900 ka interval is highlighted by pink shades.

Benthic foraminifer $\delta^{13}$C records is a well-documented proxy in tracing ventilation history of intermediate and deep-water masses in the Pacific and Atlantic [44]. ODP Site 849 was located in the Eastern Pacific, which was dominated by the Pacific deep water (PDW) [45], and ODP Site 1090 in the South Atlantic was used to track the LCDW [46]. The inter-basinal carbon isotope gradient ($\Delta\delta^{13}$C) between Pacific Site 849 and South Atlantic Site 1090 was used to infer the ventilation history of the deep basin of the Pacific. As shown, the $\Delta\delta^{13}$C (50 pt running mean) has the same trend with the $\delta$Ce curve. In specific, during 900–800 ka, these water mass proxies show an evident downward trend, indicating an increased ventilation of the PDW. Around 700 ka, the curves show an upward trend, agreeing with the reduction trend in deep-ocean circulation [1]. It is noted that the provenance had undergone profound changes, and at the same time, a rapid increase in $\delta$Eu and titanium (TiO$_2$) indicates the induced input of hydrothermal materials from the Hawaii hotspot, and the increased hydrothermal materials may similarly contribute to the reduced seawater in the study area. Therefore, we confirmed that bottom water experienced a rapid reorganization in the central Pacific during the MPT.

## 5. Conclusions

Based on magnetostratigraphy of the sediments of core PC15, we established a chronological framework for deep-sea sediments in the Kamehameha Basin in the Quaternary. The SARs are estimated as 3–7 m/Myr and show an evident increase around the MPT, comparable with those in the West Philippine Sea and the Mariana Trench. Comparing with various paleoenvironmental proxies, the SAR shift around the MPT likely resulted from the induced weathering processes around the study area and the strengthened aeolian inputs from inner Asia, which are all correlated to the enhanced glaciation in the Northern Hemisphere. Moreover, changes in regional bottom water during the MPT were studied, showing that the Kamehameha Basin was more oxic during 900–800 ka, and then turned to a relatively anoxic condition since ~700 ka. This evidence suggests a complex reorganization in regional bottom water across the MPT. Therefore, we propose that the deep-sea sediments in the Kamehameha Basin in the central tropical Pacific record some key features in global climate changes and are worthy of further investigation in future.

**Supplementary Materials:** The following are available online at https://www.mdpi.com/article/10.3390/jmse9111254/s1, Table S1: Major and trace element concentrations of different depth in the sediment core PC15, Table S2: Rare earth element concentrations (μg/g) and characteristics parameters of different depth in the sediment core PC15.

**Author Contributions:** Conceptualization and methodology, H.W. and L.Y.; sample collection, H.W. and X.D.; formal analysis, H.W., L.Y. and G.H.; original draft preparation, H.W. and L.Y. All authors have read and agreed to the published version of the manuscript.

**Funding:** This research was funded by the Key Special Project for Introduced Talents Team of Southern Marine Science and Engineer Guangdong Laboratory (Guangzhou), grant number GML2019 ZD0106; the National Natural Science Foundation of China (42177422 and 41803062); and the COMRA Project from China Ocean Mineral Resources R&D Association, grant numbers DY135-C1-1-07 and DY135-N1-1-01.

**Institutional Review Board Statement:** Not applicable.

**Informed Consent Statement:** Not applicable.

**Data Availability Statement:** Data are available on request from the author H.W. (wanghaifeng@mail.cgs.gov.cn).

**Acknowledgments:** We are grateful to all of the onboard crew members of DY29 during the 2013 scientific expedition for collecting samples. We thank Peixin Lai and Qing Chen for analyzing clay minerals, and Piaoer Fu and Yinan Deng for their help with geochemical measurements.

**Conflicts of Interest:** The authors declare no conflict of interest.

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
