# Peer review of "Geochemical and Mineral Properties of Quaternary Deep-Sea Sediments in the Central-Tropical Pacific and Its Response to the Mid-Pleistocene Transition"

_jmse, doi:10.3390/jmse9111254_

Round 1
Reviewer 1 Report
L104
Is there a reason to heat the clay mineral to 490 degrees?
Since the Kaolinite collapses at a high temperature of 550 degrees, it is noticed that heating to about 490 degrees would cause errors in semi-quantitative analysis.
L159-171
typo (demonstate --> demonstrate)
L159-171
It was confirmed that there was no explanation for CCZ's contribution.
Originally, the CCZ explanation was a quote representing the Hydrothermal process, but this time, are the author trying to investigate it under the influence of Kyushu-palau ridge?
Author Response
Response to reviewer 1
- - L104: Is there a reason to heat the clay mineral to 490 degrees?
Since the Kaolinite collapses at a high temperature of 550 degrees, it is noticed that heating to about 490 degrees would cause errors in semi-quantitative analysis.
Reply: This method has already been examined in marine sediments in the reference [18], to exclude the potential uncertainties, each sample were measured three times.
Lines 104-107
Each sample was measured 3 times under conditions of air–drying, ethylene glycol solvation, and heating at 490°C for 2 hours at atmospheric pressure [18]. Clay minerals was identified according to the position of the (001) series of basal reflections observed on the XRD diagrams [18,19].
- L159-171: typo (demonstate --> demonstrate).
Reply: agree and revised accordingly.
- L159-171: It was confirmed that there was no explanation for CCZ's contribution.
Originally, the CCZ explanation was a quote representing the Hydrothermal process, but this time, are the author trying to investigate it under the influence of Kyushu-palau ridge?
Reply: agree and revised. We add more discussion about the clay minerals to highlight the relationship between samples from various study areas.
Line 169-185
Illite is the dominant clay mineral in Chinese loesses, up to 65%, and chlorite, ~20%. Kaolinite was commonly found in loesses, about 10%, while smectite was only ~5% [21]. In the West Philippine Basin, changes in clay minerals are large: smectite is high whose content is 60%-80%, and illite and chlorite range from 16% to 37% [23-24]. In the sediments of the Mariana Trough, more smectite was reported due to regional hydrothermal and volcanic activities [25].
Although the Kamehameha Basin is distant from the Philippine Sea, the provenance between them is similar [27-28]. For example, samples from Unit II of core PC15 displace in the area of the triangular plot similar to ones of the West Philippine Basin and the Mariana Trough, while samples from Unit I are close to samples from the the Mariana Trough and the Luzon Islands, which can be correlated to the increased contribution of hydrothermal materials and volcanic debris [12]. Thus, the comparison between clay minerals indicates that aeolian deposits in the sediments generally decrease along with the distance away from the Chinese loess plateau, and hydrothermal materials and volcanic debris significantly increase. For stratigraphic changes, smectite content in the top 0.92 m (Unit I) is higher than the lower part (Unit II), illite reduces from ~30% to a much lower level, and changes in chlorite and kaolinite are small (Figure 6).
Reviewer 2 Report
The manuscript entitled “Geochemical and mineral properties of Quaternary deep-sea sediments in the central-tropical Pacific and its response to the Mid-Pleistocene transition” by Wang et al. represents a significant contribution to the paleoceanography of central tropical Pacific Ocean during the Mid Pleistocene transition. I found the topic of the manuscript very interesting and possibly of great interest for the Journal of Marine Science and Engineering readers. The manuscript is of great interest, as it examines for the first time such a synthetic view examining all these deep-sea characteristics.
Overall, all data are sufficient, and the treatment of the data are appropriated. The figures are appropriated as both quantity and quality. The length of this review paper is appropriated for this journal, with all interpretations and conclusions to be in general very well justified. The text is very well organized, and this makes the manuscript easily readable and understandable. The bibliography is accurate too. The English is in relatively good shape, but some places need some improvements (please see my comments about rephrasing below). However, there are some critical points that need to be clarified and/or better discussed before acceptance. Therefore, I propose several points to be addressed before it can be considered for publication (minor revision). Both the minor comments and suggestions are listed right below.
Minor comments and suggestions:
-L46: …and their characteristics can be employed to reveal these information
-L48-51: Split it into 2 different sentences or even better the water masses info should be removed to the Material and Methods section
-L51: This is an overstatement. Rephrase it. It is not unique, maybe ideal for the purpose of this work
-L59: Add the name of the studied core
-L58-64: The main goal of this study is not well presented here. Try to make it more clear for the potential reader
-L75-76: Add the depth of the CCD in this area and also a reference is needed for that
-L77: According to the lithological description,
-L96: Do the 28 samples cover the entire core length? From which intervals do these samples come from? Such information is crucial and should be added at this section
-L136: As particularly shown in Figure 4, …..
Section 3.2.: The relationship between hydrothermal and aeolian deposits should be better explained in that section. In more detail!
-L185: A space is needed between “variation” and “can”
-L200-202: References are missing for supporting this
-L236: Pacific and Atlantic water masses. Moreover, is there any evidence of biogenic barites?
Author Response
Response to reviewer 2
- -L46: …and their characteristics can be employed to reveal these information.
Reply: agree and revised accordingly.
- -L48-51: Split it into 2 different sentences or even better the water masses info should be removed to the Material and Methods section.
Reply: agree and revised. This sentence has been split.
Line 48-50
The Kamehameha Basin connects the Central Pacific and the Eastern Pacific (Figure 1). The Lower Circumpolar deep water (LCDW) and the Antarctic bottom water (AABW) flowed into the open Eastern Pacific Basin through the Horizon and Clarion Passages [10].
- -L51: This is an overstatement. Rephrase it. It is not unique, maybe ideal for the purpose of this work.
Reply: agree and revised.
Line 51
This region is an area with significant interactions between various systems:
- -L59: Add the name of the studied core.
Reply: agree and revised accordingly.
- -L58-64: The main goal of this study is not well presented here. Try to make it more clear for the potential reader
Reply: agree and revised.
Lines 58-65
However, such studies on the nonfossilliferous sediments were less studied in previous researches due to age uncertainties [11, 13], limiting our understanding of how deep ocean attended in global changes. In this work, we studied a gravity core PC15 collected from the Kamehameha Basin. Combining the results of magnetostratigraphy, clay minerals, and geochemistry, the properties of the deep-sea sediments were well studied in the Quaternary. By comparing the results of the core with various paleoenvironmental proxies during the MPT, we attempt to study (1) the sedimentary processes, (2) changes in the provenance, and (3) bottom water evolution in the study area.
- -L75-76: Add the depth of the CCD in this area and also a reference is needed for that.
Reply: agree and revised accordingly.
- -L77: According to the lithological description,
Reply: agree and revised accordingly.
- -L96: Do the 28 samples cover the entire core length? From which intervals do these samples come from? Such information is crucial and should be added at this section
Reply: Yes, 28 samples cover the entire core. We revised the statement.
Lines 98-99
A total of 28 samples were taken for clay mineral analysis by X-ray diffraction (XRD) at a sampling interval of 20 cm continuously from the top to the bottom of core PC15.
- -L136: As particularly shown in Figure 4, …..
Reply: agree and revised accordingly.
- Section 3.2.: The relationship between hydrothermal and aeolian deposits should be better explained in that section. In more detail!
Reply: agree and revised. More discussion was added in this version.
Lines 169-185
Illite is the dominant clay mineral in Chinese loesses, up to 65%, and chlorite, ~20%. Kaolinite was commonly found in loesses, about 10%, while smectite was only ~5% [21]. In the West Philippine Basin, changes in clay minerals are large: smectite is high whose content is 60%-80%, and illite and chlorite range from 16% to 37% [23-24]. In the sediments of the Mariana Trough, more smectite was reported due to regional hydrothermal and volcanic activities [25].
Although the Kamehameha Basin is distant from the Philippine Sea, the provenance between them is similar [27-28]. For example, samples from Unit II of core PC15 displace in the area of the triangular plot similar to ones of the West Philippine Basin and the Mariana Trough, while samples from Unit I are close to samples from the Mariana Trough and the Luzon Islands, which can be correlated to the increased contribution of hydrothermal materials and volcanic debris [12]. Thus, the comparison between clay minerals indicates that aeolian deposits in the sediments generally decrease along with the distance away from the Chinese loess plateau, and hydrothermal materials and volcanic debris significantly increase. For stratigraphic changes, smectite content in the top 0.92 m (Unit I) is higher than the lower part (Unit II), illite reduces from ~30% to a much lower level, and changes in chlorite and kaolinite are small (Figure 6).
- -L185: A space is needed between “variation” and “can”
Reply: agree and revised accordingly.
- -L200-202: References are missing for supporting this√
Reply: agree and added.
Line 212-216
Within the MPT, SARs of core PC15 and ODP Site 842 [12] significantly increased, from 3 m/Ma to 6.38 m/Ma in core PC15, and from 4.76 m/Ma to 10.63 m/Ma at ODP Site 842 [12]. This increase in SARs is not only evident in the Kamehameha Basin, but also in the southern slope of Mariana Trench [13], the western Pacific, and the Philippian Sea [30-31] (Figure 8).
- -L236: Pacific and Atlantic water masses. Moreover, is there any evidence of biogenic barites?
Reply: agree and revised accordingly. Since no direct evidence was found in this work for biogenic barites, we removed the word “biogenic” from the rest part.